# Influence of Different Litter Regimens on Ceca Microbiota Profiles in *Salmonella*-Challenged Broiler Chicks

**DOI:** 10.3390/ani15142039

**Published:** 2025-07-11

**Authors:** Deji A. Ekunseitan, Scott H. Harrison, Ibukun M. Ogunade, Yewande O. Fasina

**Affiliations:** 1Department of Animal Sciences, North Carolina Agricultural and Technical State University, Greensboro, NC 27411, USA; daekunseitan@ncat.edu; 2Department of Biology, North Carolina Agricultural and Technical State University, Greensboro, NC 27411, USA; 3Division of Animal and Nutritional Science, West Virginia University, Morgantown, WV 26506, USA; ibukun.ogunade@mail.wvu.edu

**Keywords:** 16S rRNA, microbiota, *Salmonella*, microbiome–host interaction, broiler chicks, poultry

## Abstract

Poultry litter is a mixture of bedding materials and excreta of birds, resulting in a complex biological system of microbes making up its microbial population. Its management is a critical component in poultry production, with implications for the health and productivity of the birds. This study investigated litter type (dirty litter, DL; fresh litter, FL) and *Salmonella* Enteritidis on the microbiota composition of neonate broiler chicks. This study demonstrated that dirty litter with challenge promotes beneficial microbial diversity, enhances feed intake, and supports pathogen exclusion, indicating that strategic litter management can improve chick health and performance outcomes.

## 1. Introduction

The poultry industry in the United States is one of the largest food industries worldwide, and chickens are commonly reared for meat production [1]. Annually, this industry generates over 10 million tons of litter to accommodate the growing number of birds necessary to meet consumer demands [2]. The poultry industry is faced with numerous challenges, the focal point being the issue of health. Several authors have posited diverse factors that impact the composition and diversity of microbiota in the gut of chickens. These include the age of the bird [3,4], feed additives [5], antibiotics used [6,7], and litter management [8,9,10]. Overall poultry health and growth performance are closely related to the establishment and development of a healthy and balanced gut microbiome in poultry [11].

The performance of chickens is determined not only by breeding and feeding, but also by the management conditions of the poultry unit. Beyond genetics and nutrition, environmental factors and litter management significantly influence broiler growth and carcass quality [8]. Litter management has been one of the most crucial aspects of floor-rearing systems, providing thermal insulation and promoting comfort for the birds. Litter influences broiler productivity and meat quality by absorbing moisture from excreta and maintaining bedding integrity to ensure bird comfort. The litter type in which broiler chickens are raised plays a significant role in pre-harvest health conditions, and studies have shown that flock performance is related to the quality of the litter material that the birds are raised on [9,10]. Poultry litter consists of a mixture of bedding materials and bird excreta, forming a complex biological system rich in microbial communities. The composition and structure of the poultry litter is dependent on the bedding material and on the litter management regimen used in the system [12]. Bedding can be reused for multiple successive lots when kept in an adequately dry environment and under good management, which can lower costs and environmental impact [13]. The reuse of bedding is a common practice in some developed countries like Australia, the USA, and Brazil, and sometimes occasioned by existing climatic conditions and cost implications [14]. The reused litter carries a mixed population of bacteria, fungi, and viruses. Pathogenic bacteria such as *Salmonella* spp., *Campylobacter* spp., *Escherichia coli*, *Clostridium perfringens*, and *Staphylococcus aureus* proliferate in litter, composed of substrate and the excrement of birds. These microbes, normally found in the intestines, are harbored in reused litter, which is not the case with fresh litter [15]. Chicks raised on reused litter versus fresh litter exhibited notable differences in the bacterial communities present in both the litter and their ceca [12]. The understanding of the reciprocal impact between litter microbiota and chicken GI microbiota is important to guide proper management of poultry litter and the health of birds.

A key strategy in animal health management involves limiting the proliferation of harmful and antimicrobial-resistant bacteria, both within the poultry gut and across the food production chain [16]. It is well known that gut microbiota influence chicken growth performance, with a resultant effect on the overall meat quality obtained and consumed by the population. Therefore, the management of litter is of great importance in poultry production, as it plays a major role in the accumulated population of gut microbes because there may exist an inverse relationship between the gut microbiota and litter microbes. Roll et al. [17] demonstrated that using used litter reduces the number of *Salmonella* bacteria that can infect chickens. Additionally, the conditions of the litter can have an indirect effect on the gut microbiota of chickens by way of their immune systems; moreover, the humoral and cell-mediated immune responses is stimulated when chicks are raised on reused poultry litter [18,19]. Any alteration in the intestinal microbiome may have functional consequences to the health of the host, since the intestines of broiler chickens contain a complicated microbiota that contributes greatly to the birds’ growth and overall health. A notion known as “competitive exclusion” was born out of a groundbreaking study that revealed that a healthy gut microbiome could protect broiler chicks against enteric infections [2]. This study moves beyond risk assessment to highlight the health-promoting potential of reused litter under a challenged model, aiming to bridge the gap between pathogen control (*Salmonella* management) and the science of microbiome–host interaction, positioning litter management as a tool for both food safety and animal health. Therefore, we hypothesized that experimental settings for simulating field conditions of exposing young chicks to used litter would accelerate the development of microbiota composition against enteric pathogens.

## 2. Materials and Methods

### 2.1. Description of Salmonella Strain Used for Experimentation

In this study, broiler chicks were challenged with nalidixic-acid-resistant *Salmonella* Enteritidis (SE) on d 1. The resistant mutant strain of SE resistant to nalidixic acid was generated by plating and sequential culture on xylose lysine tergitol-4 (XLT4) media containing 50 μg/mL of nalidixic acid (MP Biomedicals, Irvine, CA, USA), following established protocols [20,21]. To ensure selective recovery of the challenge strain, all culture media used for SE isolation were consistently supplemented with 50 μg/mL nalidixic acid, effectively suppressing competing microbiota while permitting the exclusive growth of the marked SE strain.

### 2.2. Experimental Design, Diet, and Bird Management

Day-old Ross 708 broiler male chicks (*n* = 260), hatched from eggs laid by 32-week-old broiler breeders, were obtained from a commercial hatchery and transported to the Poultry Research Unit at North Carolina A&T State University, Greensboro. Upon arrival, twenty (20) chicks were randomly selected and euthanized by CO_2_ asphyxiation to verify the absence of nalidixic-acid-resistant *Salmonella* Enteritidis (SE). Cecal lobes were aseptically collected into sterile Whirl-Pak^®^ bags (Whirl-Pak, WI, USA), into which 25 mL of buffered peptone water (BPW; Thermo Scientific, Hampshire, UK) was added and then homogenized at 230 rpm for 60 s using a Stomacher 80 Microbiomaster (CamLab, Cambridge, UK). The homogenates were incubated overnight at 37 °C for pre-enrichment, after which 1 mL aliquots were transferred to both tetrathionate (TT) and Rappaport–Vassiliadis (RV; Remel Inc., Lenexa, KS, USA) broths, respectively, and incubated at 42 °C for 24 h. Following enrichment, 10 μL loopful of each enriched culture was streaked onto xylose lysine tergitol-4 (XLT4; Becton, Dickinson and Company, Sparks, MD, USA) agar plates containing 50 μg/mL of nalidixic acid and incubated for 48 h at 37 °C. Thereafter, suspected SE colonies were selected and subjected to biochemical confirmation tests by inoculation onto triple sugar iron (TSI) and lysine iron agar (LIA) slants (Remel Inc., Lenexa, KS, USA) to assess characteristic fermentation and hydrogen sulfide production, as outlined by the USDA Food Safety and Inspection Service Laboratory Guide [22]. Following biochemical confirmation, *Salmonella* isolates were further analyzed using a latex agglutination test (serological verification) employing polyvalent O antiserum reactive with serogroup (A-I + Vi) antigens to facilitate serogroup-level identification of the isolates [23].

Two hundred and forty chicks were randomly assigned based on body weight to four (4) treatments in a 2 (**Litter type:** dirty litter vs. fresh litter) and 2 (***Salmonella* challenge**: nonchallenged CON, **NC** versus challenged CON-SE, **SE**) factorial arrangement. The litter regimen consisted of two types: the dirty litter (**DL**) type, which had been recycled 4 times from a commercial flock, and a fresh litter (**FL**) type. Treatment **CONDL** consisted of chicks raised on dirty litter. Treatment **CONFL** consisted of chicks raised on fresh litter. Treatments **CONDLSE** and **CONFLSE**, consisted of chicks raised on litter type regimens similar to **CONDL** and **CONFL**, respectively, but each orally inoculated with 7.46 × 10^8^ colony-forming units (CFU) SE/mL at 1 d of age. To minimize the risk of cross-contamination during *Salmonella* Enteritidis (SE) inoculation and subsequent sample collection, strict sanitary protocols were implemented throughout this study. During oral gavage, each chick was inoculated individually using a sterile syringe, discarded after single use. The procedure was conducted in a designated biosafety section, and personnel wore full personal protective equipment (PPE) and face masks. Gloves were changed between the treatment groups

Each treatment consisted of four replicate floor pens, with 15 chicks housed per pen. The pens were fitted with a hanging feeder, a nipple drinker line, and with the designated litter type. The environmental pen temperature was maintained at 33.33 °C from day 1 to day 7, and was adjusted to 30.6 °C from day 8 to day 14. A continuous photoperiod of 23 h light and 1 h dark (**23L:1D**) was applied throughout the 14-day experimental period, with the light intensity set at 30 lux for the duration of the study. Th experimental diets were formulated to meet the recommendations of the National Research Council [24].

The starter diet was fed as crumbles to the chicks throughout the 14-day experimental period. The chicks had unrestricted access to both feed and water throughout the duration of this study. The nutrient composition of the experimental diets is detailed in Table 1.

### 2.3. Preparation of Bacterial Inoculum and Salmonella Challenge

The frozen SE stock culture was surface-thawed, and 10 µL was transferred and inoculated into 10 mL of sterile tryptic soy broth (TSB, MP Biomedicals, Irvine, CA, USA). The inoculated broth containing the bacteria (SE) was incubated at 37 °C (Thermo Scientific Heratherm Advanced Protocol Microbiological Incubator, Waltham, MA, USA) overnight and streaked onto XLT4 agar plates (Becton, Dickinson and Company, Sparks, MD, USA), onto which 0.1% nalidixic acid solution (50 µL/mL) had been added during preparation. The streaked plates were then incubated at 37 °C for 48 h. Following 48 h of incubation at 37 °C, the streaked plates were examined, and a presumptive black colony of *Salmonella* Enteritidis (SE) was selected and inoculated into 10 mL of sterile tryptic soy broth (TSB). The culture was incubated at 37 °C for 24 h. This resultant culture served as the basis for preparing the challenge inoculum. The SE suspension was subsequently diluted in sterile buffered peptone water (BPW; Thermo Scientific, Waltham, MA, USA) to achieve a concentration of 7.5 × 10^8^ CFU/mL. The bacterial load of the inoculum was quantified by measuring absorbance at 687 nm using an AccuSkan Go microplate reader (Thermo Fisher Scientific, Vantaa, Finland), with reference to a standard curve generated for SE. The concentration of viable SE cells in the inoculum was then determined by streaking 10 µL onto an XLT4 plate and counting black colonies after incubating the plate overnight at 37 °C. The results showed that SE inoculum contained 7.5 × 10^8^ CFU/mL. Day-old chicks in treatments CONDLSE and CONFLSE were inoculated with the prepared SE inoculum via oral gavage of 1 mL of inoculum per chick, while each chick in CONDL and CONFL treatments were mock challenged with 1 mL of sterile BPW.

### 2.4. Chick Growth Performance Indices

Each treatment group consisted of four replicates with 15 birds at the start of the trial. On day 3 post-treatment, two birds were randomly selected and humanely euthanized from each replicate for cecal SE enumeration. Consequently, performance data were recorded from the remaining 13 birds per replicate. During performance measurements, strict sanitary procedures were followed. Disposable gloves and PPE were worn and changed between handling different groups. The instruments and weighing platforms were disinfected with 70% ethanol before and after use.

The body weight (**BW**), body weight gain (**BWG**), and feed intake (**FI**) of the chicks were measured on d 7 and d 14 for the evaluation of growth performance. The feed conversion ratio (**FCR**) was calculated from these data. Mortality was also recorded throughout the 14-day experiment.

### 2.5. Cecal Sample Collection

Cecal material was collected under aseptic conditions. All instruments used during collection were sterilized between birds. The birds were handled individually to ensure treatment integrity, and samples were immediately transferred into sterile containers and stored appropriately for downstream analyses.

For the enumeration of SE on d 3 and 14, two chicks per pen were randomly selected and euthanized via CO_2_ asphyxiation. Cecal lobes from each bird were placed into sterile Whirl-Pak^®^ filter bags (Whirl-Pak, WI, USA) containing 25 mL of sterile BPW and then homogenized at 230 rpm for 60 s using a Stomacher 80 Microbiomaster (CamLab, Cambridge, UK). Thereafter, the resulting filtrate from each sample was subjected to a ten-fold serial dilution in 10 mL of BPW (i.e., up to 10^6^), and 100 μL of each dilution was plated on XLT4 agar using the spread-plating technique. The XLT4 plates were incubated at 37 °C for 48 h. Next, the number of black presumptive SE colonies on the XLT4 agar plates was then counted for each sample. The SE concentration was expressed as log10 CFU/g ceca content.

For the analysis of ceca microbiota, three chicks were randomly selected per treatment and humanely euthanized by CO_2_ asphyxiation. Thereafter, cecal lobes were aseptically collected from each bird, placed into sterile Whirl-Pak^®^ filter bags containing 25 mL of sterile BPW, and homogenized as previously described. The resulting filtrates were subjected to DNA extraction and 16s rRNA amplicon sequencing analysis.

### 2.6. DNA Extraction

Filtrate samples were aliquoted and homogenized in 2 mL microcentrifuge tubes containing 200 mg of 106–500 μm glass beads (Sigma, St. Louis, MO, USA) and 0.6 mL of ATL lysis buffer (Qiagen, Valencia, CA, USA), supplemented with lysozyme (60 mg/mL; Thermo Fisher Scientific, Grand Island, NY, USA). The suspension was then incubated at 37 °C for 1 h with intermittent mixing. The samples were subjected to mechanical lysis via vigorous agitation at 3000 rpm for 20 min using a digital vortex mixer. Following homogenization, 600 IU of proteinase K (Qiagen) and 0.3 mL of AL buffer (Qiagen) were added, and the lysate was incubated at 70 °C for 1 h. After brief centrifugation, the clarified lysate was aspirated and transferred to a fresh tube containing 0.5 mL of absolute ethanol. The genomic DNA was purified via silica-column-based isolation, utilizing AW1 and AW2 wash buffers (Qiagen), followed by elution in nuclease-free water [25,26,27].

### 2.7. 16S rRNA Amplicon Sequencing and Bioinformatics Analysis

The amplification of the V4 region of the bacterial 16S rRNA gene was performed using 12.5 ng of total DNA and universal primers F515/R806, which included Illumina-compatible overhang adapters at the 5′ ends. Master mixes contained 12.5 ng of total DNA, 0.5 µM of each primer, and 2x KAPA HiFi HotStart ReadyMix (KAPA Biosystems, Wilmington, MA, USA). The thermal cycling protocol consisted of initial denaturation at 95 °C for 3 min, followed by 25 cycles of denaturation (95 °C, 30 s), annealing (55 °C, 30 s), and extension (72 °C, 30 s), with a final extension at 72 °C for 5 min and a 4 °C hold. Amplicons were purified using AMPure XP beads (Beckman Coulter, Indianapolis, IN, USA). A second limited-cycle PCR (8 cycles) was performed to append Illumina sequencing adapters and dual-index barcodes (i7 and i5; Illumina, San Diego, CA, USA) using the same thermal profile. The resulting libraries were purified again with AMPure XP beads, quantified, normalized, and pooled. The pooled library was denatured with NaOH, diluted in hybridization buffer, and heat-denatured prior to loading onto an Illumina MiSeq reagent cartridge. Sequencing was carried out on the MiSeq platform (Illumina) using paired-end chemistry, with automated cluster generation and sequencing performed according to the manufacturer’s protocol [28,29,30]. Raw paired-end sequencing data generated on the Illumina MiSeq PE250 platform were converted to FASTQ format and demultiplexed using Bcl2Fastq v2.18.0.12 (Illumina). The processed reads (FASTQ files) were analyzed through the Nephele microbiome analysis pipeline platform (https://nephele.niaid.nih.gov, accessed on 5 February 2025) developed by the National Institute of Allergy and Infectious Diseases (NIAID), which implemented the following steps: quality control, adapter and primer trimming of raw sequence files, and denoising and reference-based OTU clustering with a sequence similarity threshold of 97%, adhering to the default workflow settings. The OTUs were assigned to their respective taxonomic units, and final OTU counts were normalized to relative abundance (%) per sample for downstream analyses.

### 2.8. Statistical Analysis

Growth performance data were arranged in a 2 × 2 factorial layout and subjected to one-way ANOVA (Statistical Analysis Software, 2004, Version 9.2. SAS Institute Inc., Cary, NC, USA). Means separation was performed using Duncan’s multiple range test. Only data from the challenged groups (CONDLSE and CONFLSE) were analyzed for *Salmonella* Enteritidis concentration in the ceca of chicks using the *t*-test. The results were expressed as mean ± SEM, and the level of statistical significance was set at *p* < 0.05. Statistical analyses of the OTU data were conducted using the MicrobiomeAnalyst platform (https://microbiomeanalyst.ca, accessed on 20 February 2025). Sequence data were subjected to rarefaction to achieve even sampling depth (minimum library size), followed by normalization using cumulative-sum scaling (CSS). Microbial diversity was assessed through alpha diversity metrics, including richness estimators (Chao1 and ACE) and diversity indices (Shannon and Simpson), and beta diversity analysis was carried out via Bray–Curtis dissimilarity matrices, visualized through principal coordinate analysis (PCoA). Differences in diversity were evaluated using permutational multivariate analysis of variance (PERMANOVA; 999 permutations) with complementary permutational analysis of multivariate dispersions (PERMDISP). Differentially abundant taxa among the treatments were identified and visualized with comparative heat tree analysis using a non-parametric Wilcoxon rank sum test (*p* = 0.05).

## 3. Results

### 3.1. Growth Performance Parameters and Cecal SE Concentration of Chicks

The overall growth performance of the chicks from day 1 to 14 is presented in Table 2. Litter type had a significant impact on the performance indices (*p* < 0.05), except the feed conversion ratio (FCR). The birds on DL had higher body weight (0.479 kg), body weight gain (0.436 kg), and feed intake (0.685 kg) than the chickens raised on chicks on FL. For the performance indices of chickens challenged with or without *Salmonella* Enteritidis, the NC group had better (1.544 vs. 1.707, *p* = 0.0195) feed conversion ratios (FCR) than the SE group. Significant interaction effects (*p* < 0.05) were detected for FCR, FI, and mortality. SE challenge increased the FCR in the fresh litter group (CONFLSE, 1.832) but had no effects in the dirty litter group. The birds raised on dirty litter (CONDL and CONDLSE) had the highest (0.692 kg vs. 0.677 kg vs. 0.655, *p* = 0.0215) and comparable feed intake with CONFLSE. However, SE challenge reduced the mortality rate in the dirty litter group (CONDLSE; 1.667%) but not in the fresh litter group.

The establishment of SE infection was confirmed, and the SE-treated group on DL on d 3 and 14 had lower cecal SE (2.14 log 10 CFU/g and 0.89 log 10 CFU/g) compared to the SE-treated FL chicks (5.68 log 10 CFU/g and 2.48 log 10 CFU/g) (Table 3).

### 3.2. Sequencing Results and Cecal Microbial Community

The rarefaction curve (Appendix A) revealed that the number of sequences utilized for the samples were sufficient to ascertain the overall number of sequence types. The microbial community composition of the cecal samples from the treatment at the phylum level was predominantly composed of Firmicutes and Bacteroidota (Figure 1). The top ten genera revealed the presence of *Bacteroides*, *Lactobacillus*, *Faecalibacterium*, *Ruminococcus torques group*, *Clostridia UCG 014*, and *Blautia.* (Figure 2). Using four measures of alpha diversity, two measures of species richness (observed number of OTUs and Chao1), and two measures of species evenness (Shannon and Simpson indices), we identified a spectrum of diversity.

The alpha diversity of the cecal microbiota for both factors (litter type and *Salmonella* challenge) was calculated using chao1, ACE, and Simpson and Shannon indices. There was a significant difference (*p* < 0.05) in alpha diversity (Figure 3a–d) between the litter types, with higher diversity, species evenness, and richness in the dirty litter compared to the fresh litter, indicating that the use of dirty litter as a bedding material for chicks can increase the richness of the gut microbiota. However, *Salmonella* challenge exhibited no effect on species richness or evenness in the NC and SE groups (Figure 3e–h).

The beta diversity was calculated using the unweighted UniFrac distance analysis as an indication of dissimilarity in overall diversity. Permutational multivariate analysis of variance (PERMANOVA) and permutational analysis of multivariate dispersions (PERMDISP) were performed to measure whether there is significant separation of samples by litter type (DL and FL) or by SE challenge (NC and SE) due to differences in microbiome structure. Significant (*p* < 0.05) separation (PERMNOVA) was observed between the litter types, while non-significant separation was observed in the cecal microbiota data (Figure 4a,b). PERMANOVA evaluates the significance of separation between centroids of samples grouped by litter type (DL and FL) or by SE challenge (NC and SE); however, it is sensitive to multivariate dispersion, while PERMDISP determines if the distribution or spread of two sample groups are significantly different. Phylum *Tenericutes* differed between the *Salmonella*-challenged and the unchallenged groups (Appendix A). A close examination of the lineage of the major OTUs revealed that, although some genera were shared between microbiota from the fresh and the reused litter, challenged and unchallenged groups, and interaction between factors (Appendix A), some OTUs were found to be higher in one of the two litter types (*Cc_115*, *dorea*), Salmonella challenge (*Lactobacillus*, *Staphylococcus*, *Roseburia*, and *Ruminococcus*), and for interaction (*Coprococcus*, *Lactobacillus*, and *Ruminococcus*).

The comparative heat tree analysis between the litter type groups revealed (*p* < 0.05) 21 statistically different genera (Figure 5a), while 3 genera were observed between the SE and NC groups (Figure 5b). Fourteen (14) genera were highly predominant in the DL compared to FL. These genera include *Clostridia_vadinBB60_group*, *Family_XIII_AD3011_group*, *Clostridia_UCG_014*, *CHKCI001*, *Lactobacillus*, *Acetanaerobacterium*, *NK4A214_group*, *Staphylococcus*, *UCG_005*, *Lachnospira*, *Marvinbryantia*, *Corynebacterium*, *Eubacterium_ventriosum_group*, and uncultured bacteria, while seven genera, namely *Lachnospiraceae_NK4A136_group*, *Eisenbergiella*, *Ruminococcus_gauvreauii_group*, *Oscillibacter*, uncultured, *Butyricicoccus*, and *Colidextribacter*, were predominant in FL (Table 4). All three (3) genera, namely *Anaerostipes*, *Merdibacter*, and uncultured bacteria, were predominantly more abundant in NC compared to SE groups (Table 5).

## 4. Discussion

It is crucial to understand how different types of litter affect the microbiome of young chicks in order to properly manage litter and ensure the health of birds. This study used 16S rRNA gene analysis to assess the possible influence of litter regimen on the performance and gut health of young (14-day old) chicks. In the present study, DL improved the growth performance (body weight, body weight gain, and FCR) of growing chicks compared to FL chicks. This aligns with the report of [31], showing that reused litter results in performance improvements when compared to birds raised on fresh litter. This outcome can be attributed to differences in litter characteristics, microbiome composition, pH, moisture, and the amount of recycled nutrients from the previous flock. Hussien [32] reported attaining the best bird weight and 6% bird weight gain in birds raised on reused litter than those raised on fresh litter. The increased feed intake in broiler chickens raised on DL has been attributed to the beneficial synthesis of specific B-complex vitamins present in the used litter because of microbiological breakdown [33]. The poultry gut microbiome synthesizes B-complex vitamins, but limited absorption in the cecum leads to their excretion in feces; therefore, coprophagic behavior allows birds to recover these nutrients, as evidenced by higher vitamin requirements in cage-raised chickens compared to those on litter floors [11]. The early introduction of newly hatched chicks to reused or dirty litter also promotes the colonization and interaction of microbiomes between the chicks’ gut and litter, allowing them to pick up beneficial probiotic bacteria, such as *Lactobacillus*, resulting in enhanced intestinal microbiota and consequently improved performance [2,31,32,34]. Depression in performance indices such as FCR by SE challenge in CONFLSE is expected, as neonate chicks are susceptible to infection, resulting in malabsorption, impaired growth rate, inefficient feed utilization, and mortality [35,36].

The continual reduction in the *Salmonella* Enteritidis concentration in the ceca is due to the presence of a diverse microbiota population in the litter picked up by young chicks, resulting in the selective exclusion of pathogenic bacteria in chicks challenged with *Salmonella* Enteritidis and raised on DL. Dirty litter, or reused litter, has been established to inhibit the colonization of *Salmonella* [17] and *Clostridium perfringens* [6] in birds; therefore, the presence of *Salmonella* in the litter of broilers decreases as the number of flocks raised on it increases. One possible explanation for this is that reusing litter hinders the growth of pathogenic bacteria. The chicks raised on reused litter exhibited higher levels of volatile fatty acids in the cecum, resulting in enhanced resistance to *Salmonella* colonization compared to a new litter regimen [37]. According to research on the competitive exclusion of reused litter, *Salmonella* is less likely to survive in recycled litter than in fresh litter, with the likelihood of horizontal *Salmonella* transmission being lower in birds raised on used litter [17,38].

At the phylum level, the microbial community composition of the cecal samples from the treatments was predominantly composed of *Firmicutes* and *Bacteroidota*. *Firmicutes* and *Bacteroidota* have been reported to be the most dominant representative phyla and account for more than 90% of total cecal microbiota in chicks and healthy adults [4,34]. The relative abundance of *Firmicutes* has been posited to increase with chick age, attributable to the fact that it is the largest bacterial component of the gut microbiome of chicks, while the relative abundance of *Bacteroidetes* and *Proteobacteria* decreases with age [3]. *Firmicutes* is involved in the metabolism of carbohydrates, whereas *Bacteroidetes* is involved in the transportation and metabolization of amino acids and energy production [39,40]. The ratio of *Firmicutes* to *Bacteroidetes* abundance in the gut has been linked to the efficiency of energy harvesting [41]. The cecal microbiota genera of *Escherichia-Shigella*, *Ruminococcus_torques_group*, and *Lactobacillus* have been identified as key contributors, with the strongest influence on predicting intramuscular fat content in broiler thigh meat [42]. The cecal digesta of young chickens raised on reused litter under SE showed a higher prevalence of the fecal bacteria *Blautia*, *Faecalibacterium*, and *Anaerotruncus*, suggesting that reused litter may accelerate the colonization of the gastrointestinal (GI) tract, as early bacterial establishment potentially enhances colonization resistance, offering young chicks a stronger defense against pathogenic microbes. *Lactobacillus* was more abundant in the ceca of chicks raised on DL in both NC and SE. Beneficial bacteria like *Lactobacillus* thrive in the nutrient-rich environment created by the accumulation of organic matter in reused litter, unlike fresh litter, and could easily colonize the intestine of young chicks [34]. Collectively, it results in a decrease in harmful bacteria such as *Salmonella*, promoting the maintenance of immunological intestinal homeostasis and resulting in better growth performance of broilers [11,19]. In addition, the abundance of *Lactobacillus* helps in the degradation of lipids and lactose, the enhancement of goblet cell counts [43], and the enhancement of intestinal immune function through an increase in the expression levels of differential cytokine expression in the T cells of chicken cecal tonsils, resulting in intestinal homeostasis [44]. The alpha diversity was higher in both the challenged and unchallenged birds raised on dirty litter (CONDL and CONDLSE) compared to other groups raised on fresh litter (CONFL and CONFLSE), indicating that the use of dirty litter as a bedding material for chicks can increase the richness of the gut microbiota. The reuse of poultry litter has been established to alter the microbial community in the litter, which may in turn affect the chicken gut microbiota [11], as the cecal contents of young chicks raised on dirty litter (DL) has been discovered to have greater diversity compared to that of older ones raised in the same living conditions [45]. In addition, the beta diversity of the microbial community showed that the bacterial composition was different among the litter types, indicating that the bacterial communities were greatly affected by the litter regimen and could possibly change based on the age of the birds. This early microbial exposure can prime the immune system without causing disease, leading to faster immune maturation and reduced energy expenditure on chronic immune activation later in life [46]. The DL microbiome composition could account for the birds’ microbiota changes, with a mature and complex microbial community being established early in life. The genus Dorea OTUs was more abundant in the unchallenged chicks and is known to improve chickens’ feed conversion rate [47]. This aligns with the better FCR observed in unchallenged chicks in this study. However, an infection or colonization by pathogens in the intestine regulates its abundance, as observed with the higher FCR in the SE-challenged group [44,48]. *Cc_115* belongs to *Erysipelotrichaceae*, and it is involved in the digestibility of lipids and the formation of short-chain fatty acids [49]. The presence of *Cc_115* is part of a complex ecosystem in the poultry gut, which contributes positively to maintaining health, facilitating digestion, and influencing feed efficiency. Changes in its abundance can, therefore, lead to shifts in the overall microbial balance, impacting the host’s health and productivity, as observed in the performance indices of the challenged group [50,51]. These results suggest the impact of the litter management regimens and *Salmonella* infection on the cecal microbiota in young birds at the genus level.

The heat tree comparison analysis elicited some important variations in the abundance of bacteria genera in litter type and SE-challenged scenarios. The two prominent genera identified in the comparison between the SE-challenged groups were *Merdibacter* (family *Erysipeotrichaceae*) and *Anaerostipes* (*Lachnospiraceae*), which were predominant in NC. *Merdibacter* is generally regarded as a beneficial bacterial that has a significant influence on healthy microbiota balance, thereby contributing to the overall diversity of microbes [52]. The *Merdibacter* genus is known to play a key role in energy and nutrient harvesting within the gut, thereby maintaining metabolic homeostasis within the gut [53]; in addition, its abundance has been associated with microbial metabolites [54], which inadvertently results in better integrity of the intestinal barrier. *Anaerostipes* is adjudged to be an important bacterium in host health; moreover, it is a Gram-positive anaerobic genus from the phylum *Firmicutes* that enhances bone development [55] and also metabolizes carbohydrates, producing butyrate (short-chain fatty acids, SCFAs), which plays a significant role in gut stability [56]. The challenge with SE did not induce a collapse or comparative distinction of the intestinal microbiome in the current study, contrary to the report by Gillis et al. [57], stating that *Salmonella* Typhimurium infection resulted in intestinal microbiota disruption and gradual decimation of clostridia bacteria, accompanied by an alteration of butyrate and lactate levels utilized by *S.* Typhimurium for invasion establishment.

The fourteen genera were highly predominant in the DL compared to FL. These genera include *Clostridia_vadinBB60_group*, *Family_XIII_AD3011_group*, *Clostridia_UCG_014*, *CHKCI001*, *Lactobacillus*, *Acetanaerobacterium*, *NK4A214_group*, *Staphylococcus*, *UCG_005*, *Lachnospira*, *Marvinbryantia*, *Corynebacterium*, *Eubacterium_ventriosum_group*, and uncultured bacteria; while seven genera, namely *Lachnospiraceae_NK4A136_group*, *Eisenbergiella*, *Ruminococcus_gauvreauii_group*, *Oscillibacter*, *Butyricicoccus*, and *Colidextribacter* were predominant in FL. The 14 genera observed in the birds raised on DL identified via heat tree analysis clearly revealed there are distinct differences in microbial community structure between litter types. The *Lactobacillus* genera is opined to be affected by litter regimen, and its presence is favored in DL, potentially affecting host health and feed conversion efficiency [12], as observed by the higher body weight, weight gain, and better feed to gain ratio observed in the current study. The outcome contradicts the report of [2], stating that chickens raised on FL revealed an increasing colonization with *Lactobacillus* spp. The shift observed reflects the microbial contributions of previous flocks, as the reused litter’s microbiome contains more bacteria of intestinal origin, such as *Lactobacillus* [58]. *Clostridiales_vadinBB60_group* is an SCFA-producing bacterial group has been established to be positively correlated with feed efficiency [59] and also as a biomarker for enhanced performance in broilers and fat deposition [42]. This fuels the growing evidence regarding the gut–muscle axis, suggesting that muscle metabolism can be improved or modified through the regulation of gut microbiota [42,60]. *Lachnospira* is identified as an important genus within the gut microbiota of poultry, playing a noteworthy role in maintaining gut health and contributing to overall bird well-being. Studies indicate that the reuse of litter can lead to a reduction in the prevalence of antibiotic-resistant pathogens, likely due to the exclusion by beneficial bacteria, such as *Lachnospira*, against pathogenic ones, such as Salmonella. The presence of *Lachnospira* has been positively associated with an improved feed conversion ratio [61] and body weight gain in broilers [62]. The aerobic atmosphere of poultry farms and farmhouses is known to contain spores of *Clostridiales*, families of *Lachnospiraceae* and *Ruminococcaceae*, which explains the dominance of *Lachnospira* in the DL [63]. The concomitant effect is observed in the significant positive changes in performance and *Salmonella* concentrations in DL chicks. *Lachnospira* possess anaerobic and fermentative abilities and are capable of degrading non-starch polysaccharides, thereby producing butyrate [64], the primary energy needed for epithelial cell growth, barrier integrity, and the inhibition of inflammatory responses. Reused litter (DL) has been associated with the genera *Yaniella*, *Staphylococcus*, *Brevibacterium*, and *Salinicoccus* [12], consistent with the observed colony population of Staphylococcus in the ceca of the young chicks on DL. *Clostridia UCG-014* has been reported to be positively associated with the production of SCFA (n-butyric acid) [64], bacterial diversity, and barrier function—especially the improvement of the gut barrier function through the activation of tryptophan metabolism—while *Oscillospiraceae UCG_005* has been reported to be positively correlated with SCFA concentration [65]. In addition, an increased abundance of beneficial genera such as *Oscillospirales UCG-005*, *Roseburia*, and *Bifidobacterium* cumulates in an improved gain in weight, as observed in the birds on DL for body weight and body weight gain compared to those on FL [66]. An increased presence of beneficial bacteria such as *Lachnospiraceae_NK4A214_group* and *Lactobacillus* can help to preserve the balance of the microbial structure, with the former being directly correlated with isobutyrate and isovalerate concentrations needed for anti-inflammatory properties, potentially contributing to maintaining gut homeostasis by modulating immune responses [67]. *Lactobacillus* is known to positively influence villus height and the villus height:crypt depth ratio, thereby providing a larger surface area and a greater ability of nutrient absorption, resulting in the improved FCR and performance of broilers observed in DL [68]. These findings imply that DL is beneficial to the microbial community and enhances gut health by modulating microbiota functions. The growing population of *Marvinbryantia* in the cecal microbiota of chicks on DL reflects a beneficial gut microbiota profile, as it is known to have anti-inflammatory effects and aid in mucosal regeneration [69]. This is supported by the SE concentration reduction in DL, counteracting the excessive inflammatory response triggered by SE infection and essentially mitigating the damage caused by the bacteria through the consumption of gut mucus. This results in an impaired mucus barrier, thereby allowing microbes to reach the epithelium, cause inflammation, and exacerbate infection [70].

It is well established that the gut microbiota can influence abdominal fat deposition by modulating fat metabolism [71]. The higher abundances of *Oscillibacter*, as observed in FL at 14 days, has been documented and is predicted to result in upregulating the expression of fat anabolism genes, resulting in increased fat deposition through fat anabolism. The bacteria facilitate this process via the extraction of extra energy by utilizing CAZymes and transporting it to adipose tissues, resulting in more fat deposition [72,73]. *Eisenbergiella* is the one of the major components in bird microbiota and increased on day 7, while decreasing on day 30, and was higher in FL. Both *Butyricicoccus and Eisenbergiella* genera have been documented to be negatively correlated with villus height [68], resulting in reduced nutrient absorption and accounting for the observed poor feed to gain ratio and reduced growth indices. It is also associated with dysbiosis of the intestinal microbiota [74].

The contact time between young chicks and the litter plays a major role in the gut microbiome population and establishment [75]. This is evident in the comparison between DL and FL, and the likelihood of a chick being rapidly colonized by this anaerobe non-forming spore family is low. A longer time is typically needed for its appearance in the gut microbiota in young chicks. This may be accelerated when raised with older birds or on litter rolled over for several production cycles, as evidenced by the higher number of anaerobic bacteria groups in the DL birds. In addition, the interaction between the chicks and DL will not allow a gradual development of microbiota, rather, the chicks will acquire full or part of the litter microbiome [63].

## 5. Conclusions

In conclusion, litter type and SE challenge influenced growth performance, cecal *Salmonella* concentration, and cecal microbiota in young chicks. These findings may become increasingly important for developing alternative management strategies through litter management regimens in promoting gut health and improving poultry performance in the absence of in-feed antibiotics. Using dirty litter improved and increased the richness of the gut microbiota in terms of beneficial microbes in chicks at a younger age. This finding suggests that the use of dirty litter can be a beneficial management practice in poultry farming, enhancing chick health by promoting a diverse and robust gut microbiome and potentially mitigating the effects of pathogenic challenges like *Salmonella* Enteritidis. A more detailed gut developmental study as birds age will help us to understand the chick microbiome composition for better health management regimens, while future studies using qPCR can be utilized to determine the possible effects of litter regimens on the prevalence and abundance of *Salmonella* virulent genes.

## Figures and Tables

**Figure 1 animals-15-02039-f001:**
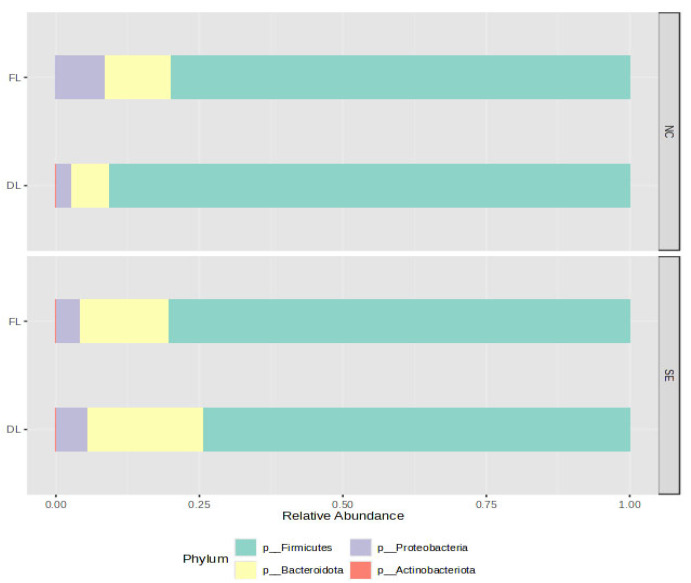
Relative abundance of microbial taxa at the phylum level. DL = Dirty litter, FL = Fresh litter, NC = nonchallenged, SE =*Salmonella* challenge.

**Figure 2 animals-15-02039-f002:**
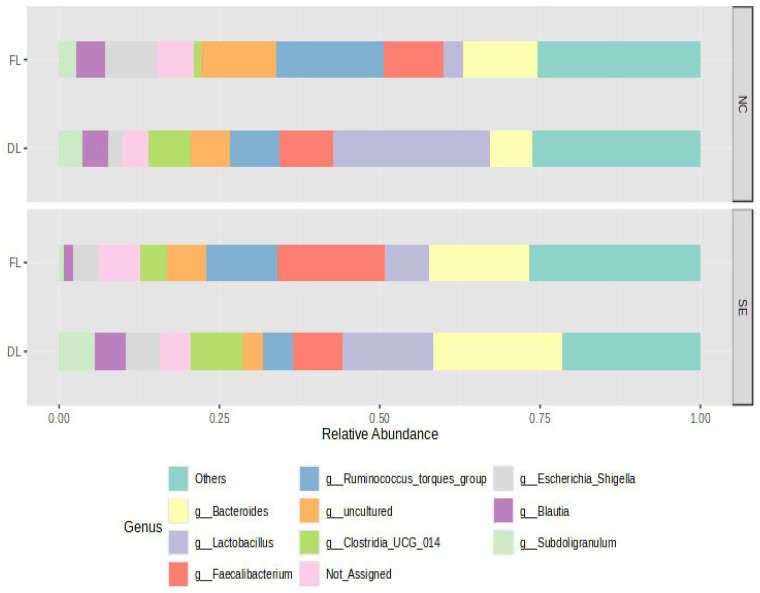
Relative abundance of the top 10 genera. DL = Dirty litter, FL = Fresh litter, NC = nonchallenged, SE =*Salmonella* challenge.

**Figure 3 animals-15-02039-f003:**
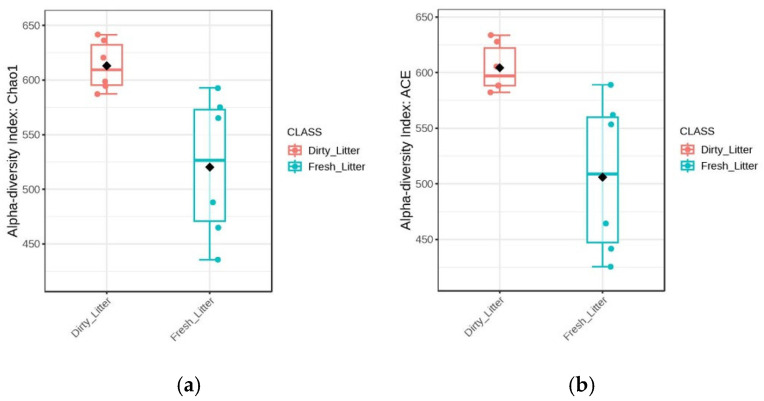
Alpha diversity indices (Chao1, ACE, Simpson, and Shannon) of microbial taxa in chicks: (**a**–**d**) litter type effect on alpha diversity index; *Salmonella* Enteritidis effect on alpha diversity (**e**–**h**). NC = nonchallenged, SE =*Salmonella* challenge.

**Figure 4 animals-15-02039-f004:**
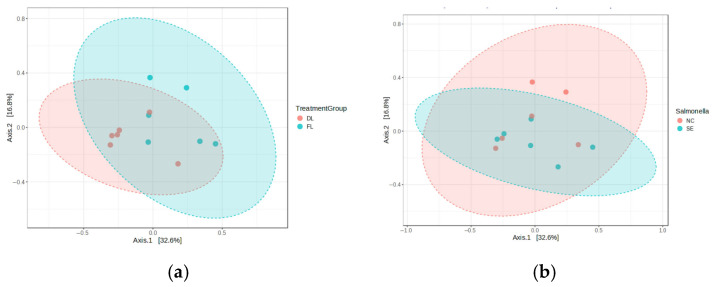
Principal coordinate analysis (PCoA) of microbiota based on Bray–Curtis index: (**a**) litter type effect (beta diversity *p* = 0.016); (**b**) *Salmonella* Enteritidis (*p* = 0.526). DL = Dirty litter, FL = Fresh litter, NC = nonchallenged, SE =*Salmonella* challenge.

**Figure 5 animals-15-02039-f005:**
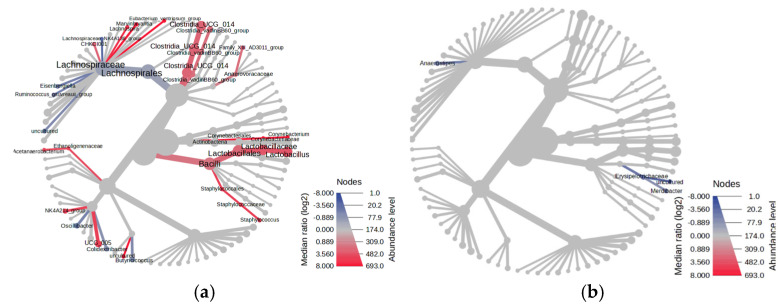
Comparative tree heatmap analysis between treatment groups (*p* < 0.05). (**a**) Dirty litter (DL) vs. fresh litter (FL), (**b**) *Salmonella* Enteritidis challenge (SE) vs. nonchallenged (NC).

**Table 1 animals-15-02039-t001:** Experimental starter diet (d 1 to 14) composition (%).

Ingredient	Control Diet
Corn (7.5% Crude protein)	51.46
Soybean meal (47.5% Crude Protein)	40.39
Poultry fat	3.64
Limestone	1.07
Mono-Dicalcium phosphate	2.03
Salt NaCl	0.40
Sodium bicarbonate	0.02
L-Lysine HCl 98%	0.13
DL-Methionine 99.0%	0.34
L-Threonine 98.5%	0.11
NCSU Poultry Vitamin Premix ^1^	0.05
NCSU Poultry Mineral Premix ^2^	0.20
Choline chloride 60%	0.10
Selenium Premix	0.05
**Analyzed nutrient composition ^3^**	
Metabolizable energy (Kcal/kg)	3117
Crude Protein, %	24.63
Crude Fat, %	4.74
Crude Fiber, %	2.3
Ash, %	6.32
**Calculated nutrient composition**	
Total Sulfur Amino Acids, %	1.03
Lysine, %	1.42
Calcium, %	0.96
Available phosphorus, %	0.48

^1^ Vitamin Premix, supplied per kilogram of diet: Vitamin A (6600 IU), Vitamin D (1980 IU), Vitamin E (33 IU), Vitamin B12 (0.02 mg), Biotin (0.13 mg), Menadione (1.98 mg), Thiamine (1.98 mg), Riboflavin (6.60 mg), d-Pantothenic Acid (11.0 mg), Vitamin B6 (3.96 mg), Niacin (55.0 mg), Folic Acid (1.1 mg). ^2^ Mineral Premix, supplied per kilogram of diet: 60 mg Manganese (Mn) as MnO; 60 mg Zinc (Zn) as ZnO; 40 mg Iron (Fe) as FeSO_4_·H_2_O; 5 mg Copper (Cu) as CuSO_4_; 1.2 mg Iodine (I) as Ca(IO_3_)_2_; 0.5 mg Cobalt (Co) as CoSO_4_·7H_2_O. ^3^ Experimental diets were analyzed for proximate nutrient composition by Eurofins Scientific Inc. Nutrient Analysis Center, 2200 Rittenhouse Street, Suite 150, Des Moines, IA 50321, USA.

**Table 2 animals-15-02039-t002:** Effect of litter type and *Salmonella* challenge on growth performance of broiler chicks from day 1 to 14.

Treatment		Body Weight (BW, kg/Bird)	Body Weight Gain (BWG, kg/Bird)	FCR (kg:kg)	Feed Intake (FI, kg/Bird)	Mortality (%)
Litter type	FL	0.420 ± 0.006 ^b^	0.363 ± 0.005 ^b^	1.682 ± 0.082	0.609 ± 0.023 ^b^	9.167 ± 2.159
	DL	0.479 ± 0.009 ^a^	0.436 ± 0.007 ^a^	1.57 ± 0.019	0.685 ± 0.013 ^a^	5.83 ± 1.967
	***p*-Value**	0.0004	<0.00001	0.0790	0.0037	0.199
*Salmonella* challenge	NC ^1^	0.448 ± 0.019	0.406 ± 0.018	1.544 ± 0.022 ^b^	0.628 ± 0.032	8.333 ± 1.667
	SE	0.45 ± 0.009	0.393 ± 0.016	1.707 ± 0.072 ^a^	0.666 ± 0.01	6.667 ± 2.520
	***p*-Value**	0.8568	0.1758	0.0195	0.0779	0.510
Litter type × SE challenge	CONFL	0.409 ± 0.006	0.368 ± 0.005	1.531 ± 0.043 ^b^	0.564 ± 0.018 ^b^	6.667 ± 2.722 ^ab^
	CONFLSE	0.431 ± 0.005	0.358 ± 0.009	1.832 ± 0.095 ^a^	0.655 ± 0.018 ^a^	11.667 ± 3.191 ^a^
	CONDL	0.488 ± 0.018	0.444 ± 0.013	1.558 ± 0.022 ^b^	0.692 ± 0.026 ^a^	10.00 ± 1.925 ^a^
	CONDLSE	0.470 ± 0.006	0.428 ± 0.006	1.581 ± 0.034 ^b^	0.677 ± 0.009 ^a^	1.667 ± 1.667 ^b^
	***p*-Value**	0.0834	0.7346	0.0375	0.0215	0.0190

^a,b^ Mean values bearing different superscript letters within a column are significantly different (*p* < 0.05). ^1^ NC = nonchallenged treatments. This represents the pooled mean of treatments in which chicks were not challenged with *Salmonella* spp.

**Table 3 animals-15-02039-t003:** Effect of litter type on the concentration of *Salmonella* Enteritidis in the ceca of broiler chickens.

	Log_10_ CFU/g Cecal Contents	
Treatment	d 3	d 14
DL	2.14 ± 0.24 ^b^	0.89 ± 0.50 ^b^
FL	5.68 ± 0.38 ^a^	2.48 ± 0.06 ^a^
***p*-value**	<0.00001	0.02

^a,b^ Mean values bearing different superscript letters within a column are significantly different (*p* < 0.05).

**Table 4 animals-15-02039-t004:** Comparative heat tree analysis of litter type on the cecal microbiota of broiler chickens.

Genus	Litter Type ^1^		Mean Diff	Wilcox *p*-Value
*Lactobacillus*	DL	FL	0.147335	0.004329
*Clostridia_vadinBB60_group*	DL	FL	0.004405	0.025974
*Lachnospira*	DL	FL	0.000647	0.012436
*Staphylococcus*	DL	FL	0.000094	0.034087
*Oscillospiraceae UCG_005*	DL	FL	0.016222	0.015152
*Lachnospiraceae_NK4A214_group*	DL	FL	0.001542	0.025974
*Marvinbryantia*	DL	FL	0.000837	0.007796
*Family_XIII_AD3011_group*	DL	FL	0.000727	0.020022
*Clostridia_UCG_014*	DL	FL	0.044653	0.025974
*CHKCI001*	DL	FL	0.002965	0.002165
*Acetanaerobacterium*	DL	FL	0.000044	0.029480
*Corynebacterium*	DL	FL	0.000281	0.002778
*Eubacterium_ventriosum_group*	DL	FL	0.000155	0.009622
Uncultured bacterium	DL	FL	0.004984	0.012436
*Oscillibacter*	DL	FL	−0.004972	0.041126
*Lachnospiraceae_NK4A136_group*	DL	FL	−0.001933	0.008658
*Eisenbergiella*	DL	FL	−0.005624	0.025974
*Butyricicoccus*	DL	FL	−0.024286	0.041126
*Ruminococcus_gauvreauii_group*	DL	FL	−0.007363	0.025974
*Colidextribacter*	DL	FL	−0.005041	0.025974
Uncultured bacterium	DL	FL	−0.053002	0.041126

^1^ DL: Dirty litter, FL: Fresh litter.

**Table 5 animals-15-02039-t005:** Comparative heat tree analysis of *Salmonella* Enteritidis on the cecal microbiota of broiler chickens.

Genus	SE Challenge ^1^		Mean Diff	Wilcox *p*-Value
*Anaerostipes*	SE	NC	−0.009722	0.004329
*Uncultured bacterium*	SE	NC	−0.004216	0.015152
*Merdibacter*	SE	NC	−0.001706	0.041126

^1^ SE: *Salmonella* Enteritidis, NC; No challenge.

## Data Availability

The original contributions presented in this study are included in the article/Appendix A. Further inquiries can be directed to the corresponding author(s).

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
