# Peer review of "Influence of Different Litter Regimens on Ceca Microbiota Profiles in Salmonella-Challenged Broiler Chicks"

_animals, 2025, doi:10.3390/ani15142039_

Round 1

Reviewer 1 Report

Comments and Suggestions for Authors

My comments are presented in the  attached file.

Author Response

Response is attached as a PDF file. 

Reviewer 2 Report

Comments and Suggestions for Authors

Ekunseitan et al. investigated the impact of different litter management practices (dirty litter and fresh litter) on the cecal microbiota of chicks under Salmonella infection challenge. The study found that dirty litter (DL) can promote beneficial microbial diversity, enhance feed intake, and support pathogen exclusion, thereby improving the health and productive performance of chicks. The composition of the cecal microbiota was also analyzed through 16S rRNA gene sequencing, revealing higher microbial diversity in the dirty litter group, with certain beneficial genera (such as Lactobacillus) being more abundant in this group. However, the article itself has several shortcomings that need to be addressed.

1 The authors are requested to carefully review the grammar throughout the manuscript. For example, lines 51-54 contain a chaotic sentence structure lacking a subject and predicate; lines 56-59 are overly verbose with repetitive information; lines 61-62 and 72-74 exhibit loose sentence structures without clear causal relationships; lines 65-66 and 83-85 are incomplete, missing subjects and predicates; lines 78-82 lack connecting words between sentences, resulting in incoherent meaning; and lines 93-95 are loose and repetitive. Similar issues will not be highlighted again in the following text.

2 L147-148 Please standardize the notation of temperature throughout the manuscript.

3 L143 What is the basis for selecting the challenge dose?

4 L155 How should interpret the term % “as is” in this context?

5 L160-161 Please specify the exact forms in which the mineral elements were added.

6 L188 on d 3 and 14, two (2) chicks were …L192 On d 14, another set of chicks (three) were …What is the basis for the selection of the number of chickens?

7 L223 Please clarify what criteria were used for filtering the raw data.

8 L235 Please explain what the method of multiple comparisons correction is.

9 L254-255 Birds on DL had higher body weight (0.479 kg), body weight gain (0.436 kg), and feed intake (0.685 kg) than chickens raised on chicks on FL.

OR:

Birds on DL had significantly higher body weight (0.479 kg vs 0.420 kg, P = 0.0004), body weight gain (0.436 kg vs 0.363 kg, P < 0.0001), and feed intake (0.685 kg vs 0.609 kg, P = 0.0037) than chickens raised on FL.

It is recommended that the entire results section be revised in accordance with this standard.

10 L255-257 It is recommended that the authors first highlight the indicators that show significant differences before discussing those that do not exhibit significant differences.

11 L346-348 Please elucidate the potential biological mechanisms by which dirty litter (DL) may enhance growth performance.

12 L353-355 This study did not detect vitamins or related microbial functional genes.

13 L357 "probiotic bacteria", please explain which bacteria they are?

14 L392 …nutrient-rich environment…Please supplement its specific content.

15 L402-403 only the litter regimen influenced the,????

16 L409 Attributing the change in beta diversity to "age" is inconsistent with the research design (only 14 days).

17 L416-420 Please supplement the references or indicate the specific functional mechanism of this genus in the chicken intestine.

18 L437-432 The author claims that the results of this study are contrary to those of others, but no comparative analysis is provided.

19 L453-455 Reference [2] shows that the abundance of Lactobacillus in the FL group is higher, which is contrary to the current result. Please supplement the reasons for the difference.

20 L480 positive gain in weight?

21 L507 higher feed to gain ratio?higher?

22 Please verify whether references 41, 46, 70, and 71 are appropriate for this manuscript.

23 Please read through the entire text and carefully check the format. Commas are missing in many places throughout the text, and even the completeness of the sentences cannot be guaranteed.

Comments on the Quality of English Language

The language logic is disordered, repetitive and wordy.

Author Response

Response is attached as a PDF file. 

Round 2

Reviewer 2 Report

Comments and Suggestions for Authors

Comments 5: L160-161 Please specify the exact forms in which the mineral elements were 
added. I mean what form is copper added in, for example, copper sulfate or copper oxide? Please supplement other corresponding mineral element addition forms.

Comments 9: ....
It is recommended that the entire results section be revised in accordance with this standard. 
Response 9: Thank you for pointing this out.  ????

Author Response

Review comment is attached
